# Application of *Pichia kudriavzevii* NBRC1279 and NBRC1664 to Simultaneous Saccharification and Fermentation for Bioethanol Production

Hironaga Akita [1] , Tetsuya Goshima [2], Toshihiro Suzuki [1], Yuya Itoiri [3], Zen-ichiro Kimura [3] and Akinori Matsushika [1,4,*]

1   Research Institute for Sustainable Chemistry, National Institute of Advanced Industrial Science and Technology (AIST), 3-11-32 Kagamiyama, Higashi-Hiroshima, Hiroshima 739-0046, Japan; h-akita@aist.go.jp (H.A.); suzuki.toshihiro222@gmail.com (T.S.)

2   National Research Institute of Brewing, 3-7-1 Kagamiyama, Higashi-Hiroshima, Hiroshima 739-0046, Japan; t.goshima@nrib.go.jp

3   Department of Civil and Environmental Engineering, National Institute of Technology, Kure College, 2-2-11 Aga-minami, Kure, Hiroshima 737-8506, Japan; yuuya.i.0830@gmail.com (Y.I.); z-kimura@kure-nct.ac.jp (Z.-i.K.)

4   Graduate School of Integrated Sciences for Life, Hiroshima University, 1-3-1 Kagamiyama, Higashi-Hiroshima, Hiroshima 739-8530, Japan

*   Correspondence: a-matsushika@hiro.kindai.ac.jp; Tel.: +81-82-493-6843

**Abstract:** Simultaneous saccharification and fermentation (SSF) is capable of performing enzymatic saccharification and fermentation for biofuel production in a single vessel. Thus, SSF has several advantages such as simplifying the manufacturing process, operating easily, and reducing energy input. Here, we describe the application of *Pichia kudriavzevii* NBRC1279 and NBRC1664 to SSF for bioethanol production. When each strain was incubated for 144 h at 35 °C with Japanese cedar particles, the highest ethanol concentrations were reached $21.9 \pm 0.50$ g/L and $23.8 \pm 3.9$ g/L, respectively. In addition, $21.6 \pm 0.29$ g/L and $21.3 \pm 0.21$ g/L of bioethanol were produced from Japanese eucalyptus particles when each strain was incubated for 144 h at 30 °C. Although previous methods require pretreatment of the source material, our method does not require pretreatment, which is an advantage for industrial use. To elucidate the different characteristics of the strains, we performed genome sequencing and genome comparison. Based on the results of the eggNOG categories and the resulting Venn diagram, the functional abilities of both strains were similar. However, strain NBRC1279 showed five retrotransposon protein genes in the draft genome sequence, which indicated that the stress tolerance of both strains is slightly different.

**Keywords:** *Pichia kudriavzevii*; NBRC strain; SSF; bioethanol; Optimash BG; Acremonium cellulase; lignocellulosic feedstocks; draft genome

## 1. Introduction

Fossil fuels are the main energy sources in the world, and demand for these fuels is growing annually. In 2015, fossil fuel consumption was estimated at 78.4% of the global final energy consumption [1]. However, the use of fossil fuels emits greenhouse gasses, which are responsible for global warming and climate change. For these reasons, biofuels that would mitigate greenhouse gas emissions have been investigated as potential renewable energy sources to replace fossil fuels. Bioethanol is one of the most widely used biofuels globally. In 2015, bioethanol was used in 66 countries, and its production reached about 25.6 billion gallons [2].

Based on the source materials or the production method, bioethanol is generally classified as first, second, or third generation. Second-generation bioethanol is produced from lignocellulosic biomass, which is widely distributed on the earth and does not raise concerns about food sustainability [3]. The microbial process for producing second-generation

bioethanol typically consists of three steps: pretreatment, enzymatic hydrolysis, and fermentation [4]. At the pretreatment step, lignocellulosic biomass is hydrolyzed into cellulose, hemicellulose and lignin by hydrothermal treatment or adding organic acids. Subsequently, at the enzymatic hydrolysis step, cellulose and hemicellulose are converted into hydrolysate containing mixed sugars. Finally, the hydrolysate is used as a carbon source in the fermentation step. However, many processes based on this method have not been put to practical use because they are not economically viable. To construct a cost-effective bioethanol production method, simultaneous saccharification and fermentation (SSF) has been studied since around the 2010s. In this method, enzymatic hydrolysis and fermentation are carried out simultaneously in the same container so that the manufacturing process can be simplified. Moreover, SSF is advantageous for its ease of operation and its reduced energy input. However, most of the enzymes used for the hydrolysis of lignocellulosic biomass show the optimum temperatures at around 50–70 °C. By contrast, the optimum temperature for fermentation (approximately 30 °C) using the budding yeast *Saccharomyces cerevisiae*, which is widely used for ethanol production, is lower than the optimal temperature for enzymatic hydrolysis. Therefore, a thermotolerant yeast is necessary to increase the productivity of bioethanol in SSF.

A thermotolerant yeast, *Pichia kudriavzevii* (previously known as *Issatchenkia orientalis*), has been isolated from various environments. *P. kudriavzevii* can catabolize a variety of different carbon sources and grows at around 45 °C as well as in high salt and sugar concentrations [5,6]. Based on those abilities, in previous studies, *P. kudriavzevii* species were applied to SSF for bioethanol production using soft biomass as the source material. For example, *P. kudriavzevii* SI is capable of producing 22.6 g/L ethanol from sugarcane bagasse [7]. Using cotton stalks, 19.8 g/L ethanol can also be produced by *P. kudriavzevii* HOP-1 [8]. In contrast, hard biomass has rigid cell walls and hence its enzymatic hydrolysis is difficult compared to soft biomass. Thus, there are relatively few reports of SSF using hard biomass. Moreover, no previous study has reported the application of NBRC strains of *P. kudriavzevii* to SSF for bioethanol production.

In the present study, we demonstrated that a combination of saccharification enzymes such as Optimash BG and Acremonium cellulase was effective for glucose production from cellulose and hemicellulose. Subsequently, we performed SSF using *P. kudriavzevii* NBRC1279 and NBRC1664 with particles from Japanese cedar or eucalyptus. To elucidate the characteristics of both strains, they were each subjected to genome sequencing and genome comparison.

## 2. Materials and Methods

### 2.1. Effect of Temperature on Enzymatic Hydrolysis of Cellulose and Hemicellulose

Using filter paper (3MM Chr; Whatman, Kent, UK) as a substrate, the optimal temperatures were evaluated by determining the glucose concentration at temperatures ranging from 30 °C to 65 °C. The reaction mixture (1 mL) contained 50 mM acetate buffer (pH 5.0), 10 FPU/L Acremonium cellulase (Meiji Seika Pharma, Tokyo, Japan), 20 µL/L Optimash BG (Genencor, Palo Alto, CA, USA), and 3MM Chr (1 cm × 1 cm). After the reaction mixture was incubated for 60 min at each temperature, the glucose concentration was determined by high performance liquid chromatography (HPLC). The determination of glucose is described in the following section. All data were obtained from at least triplicate experiments with the same enzymes.

### 2.2. Simultaneous Saccharification and Fermentation using P. kudriavzevii NBRC1279 and NBRC1664

The wood chips from Japanese cedar or eucalyptus were ground to <0.2 µm size particles by a cutter mill (MKCM-3; Masuko Sangyo, Saitama, Japan), and were dried in vacuo at 40 °C. The eucalyptus chips consisted of materials from six species (main component: *Eucalyptus globulus*). Japanese cedar woody particles were composed of 38.2% glucan, 6.0% xylan, 8.4% mannan, and 34.6% lignin. Japanese eucalyptus woody particles were

composed of 40.0% glucan, 10.4% xylan, and 28.8% lignin. These lignocellulosic biomasses were used as substrates for SSF.

Using the dried particles of Japanese cedar or eucalyptus as source material, the SSF experiments were performed. *P. kudriavzevii* NBRC1279 and NBRC1664 were separately pre-cultured aerobically for 16 h at 30 °C in YPD medium, which included 10 g/L yeast extract, 20 g/L peptone, and 20 g/L glucose. Subsequently, the pre-cultures were separately collected by centrifugation, and washed twice with sterilized water. Finally, the washed cultures were diluted to OD600 = 2.0 with 30 mL of SSF medium. SSF experiments were performed in a 50 mL screw-capped vial with a magnetic stirrer under the following conditions: 3 g/L pretreated woody particles, 5 g/L yeast extract, 5 g/L peptone, 2 g/L $NH_4Cl$, 1 g/L $KH_2PO_4$ and 0.3 g/L $MgSO_4 \cdot 7H_2O$ in 20 mM citrate buffer (pH 5.0) with 1000 FPU/L Acremonium cellulase and 2 mL/L Optimash BG at 120 rpm for 96 h at each temperature.

The effect of culture temperature on bioethanol production in SSF was examined at 30–45 °C. All data were obtained from at least triplicate experiments.

### 2.3. Quantification of Ethanol and Sugars

After each culture was clarified by centrifugation and filtration, the resultant supernatant was subjected to HPLC with an Aminex HPX-87H cationic exchange column (Bio-Rad Labs, Richmond, CA, USA). Concentrations of ethanol, glucose, and xylose were determined using a refractive index indicator. The chromatographic conditions were as follows: mobile phase, 1.5 mM $H_2SO_4$; flow rate, 0.6 mL/min; and the column oven temperature, 80 °C.

### 2.4. Genomic DNA Preparation

*P. kudriavzevii* NBRC1279 and NBRC1664 were separately cultured aerobically for 16 h at 30 °C in YPD medium. Cells were then separately harvested by centrifugation and washed twice with sterile water. Genomic DNA was extracted using the MasterPure Yeast DNA Purification Kit (Epicenter, Madison, WI, USA) and purified using AMPure XP (Beckman Coulter, Brea, CA, USA), according to the manufacturers' instructions. The concentration and purity of genomic DNA were measured by a Quant-iT dsDNA BR assay kit (Invitrogen, Waltham, MA, USA) and a NanoDrop ND-1000 spectrophotometer (Thermo Fisher Scientific, Waltham, MA, USA), respectively.

### 2.5. Genome Sequencing

To construct genome sequence libraries, genomic DNAs from *P. kudriavzevii* NBRC1279 (5 μg) and NBRC1664 (5 μg) were separately fragmented into approximately 20-kb pieces using a g-TUBE (Covaris, Brighton, UK). The resultant fragments were ligated to the sequencing adapters using a SMRTbell Template Prep Kit 1.0 (Pacific Biosciences, Menlo Park, CA, USA), yielding SMRTbell libraries. These SMRTbell libraries were selected using a BluePippin DNA Size-Selection System (15-kb size cutoff) (Sage Science, Beverly, MA, USA). The concentration and library size of SMRTbell libraries were measured using an Agilent 2200 TapeStation (Agilent Technologies, Santa Clara, CA, USA). The SMRTbell libraries were then bound to polymerases and sequencing primers using the DNA/Polymerase Binding Kit P6 v2 (Pacific Biosciences) to yield genome sequencing templates. Calculation of the concentration of the polymerase-template complex for binding and annealing reaction was performed by Binding Calculator v2.3.1.1 (Pacific Bio-sciences). Polymerase-template complexes were bound to magnetic beads using a MagBead Kit (Pacific Biosciences) and loaded on a total of 16 SMRT Cell 8 Pac v3 (Pacific Biosciences). Genome sequencing was then performed on a PacBio RS II (Pacific Biosciences).

### 2.6. Genome Assembly, Annotation, and Comparison

Raw data from the 16 SMRT Cells containing genomic DNAs of *P. kudriavzevii* NBRC1279 and NBRC1664 were assembled de novo using SMRT Analysis v2.3.0 (Pacific Biosciences;

Ref. [9]) to filter subreads and CCS reads. Prediction of the open reading frame and annotation were performed with MAKER2 v2.31.3 [10], AUGUSTUS v3.0 [11], and NCBI BLAST v2.2.29+ [12]. The tRNA gene was detected using tRNAScan-SE v1.23 [13]. The Venn diagram based on their predicted coding sequences of *P. kudriavzevii* NBRC1279 and NBRC1664 was constructed using OrthoVenn2 [14]. The protein coding sequences of *P. kudriavzevii* NBRC1279 and NBRC1664 were compared against the eggNOG database.

## 3. Results and Discussion

### 3.1. Use of Optimash BG and Acremonium Cellulase for the Enzymatic Hydrolysis of Cellulose and Hemicellulose

Optimash BG, a mixture of β-glucanase and xylanase, can hydrolyze cellobiose, cellulose and hemicellulose, respectively. Acremonium cellulase has β-glucosidase activity and catalyzes the hydrolysis of cellobiose to produce bimolecular glucose. Thus, we considered that the simultaneous use of Optimash BG and Acremonium cellulase is an effective method for glucose production from lignocellulosic feedstocks.

To enhance sugar production by enzymatic hydrolysis using Optimash BG and Acremonium cellulase, we evaluated the effect of temperature at the optimal pH (5.0) for the enzymatic activity. Filter paper 3MM Chr is made of cellulose and hemicellulose, so it was used as a substrate. When the filter paper was hydrolyzed by the enzymes, glucose was obtained as the main sugar (Figure 1). When the effect of temperature on sugar production was assessed, the glucose concentration was increased with the increase in temperature, and the concentration reached the maximum ($517 \pm 3.00$ mg/L) at 60 °C (Figure 1) with the productivity of 8.62 mg/(L·min) (Table 1).

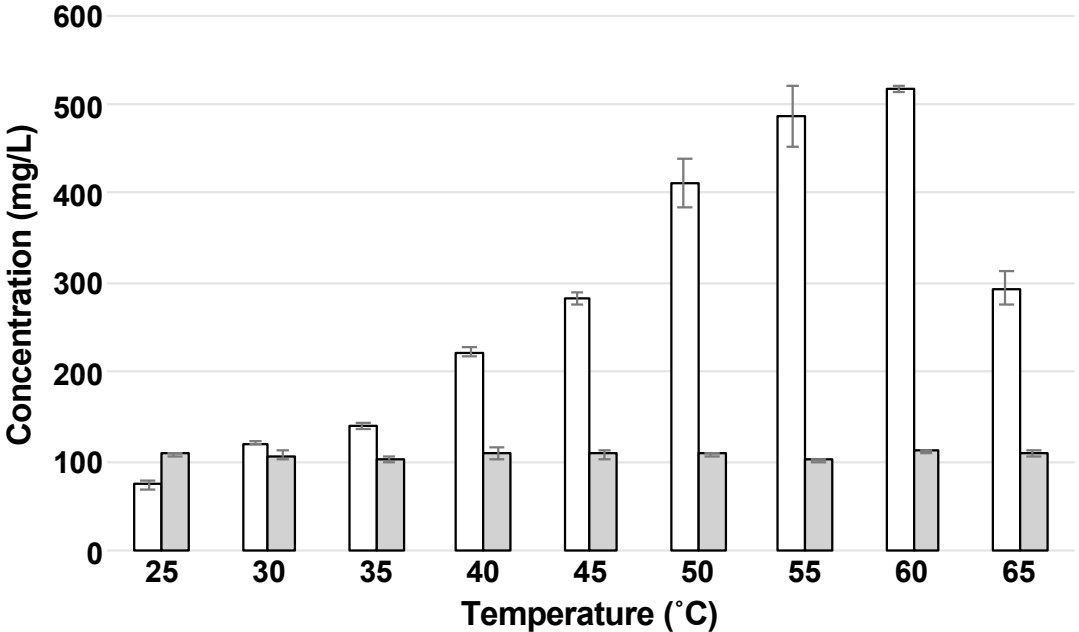

**Figure 1.** Effect of temperature on enzymatic hydrolysis. The concentrations of glucose and xylose are indicated as white and gray bars, respectively. Error bars indicate SE (*n* = 3).

**Table 1.** Glucose productivity.

| Temperature (°C) | Glucose Productivity [mg/(L·min)] |
|---|---|
| 25 | 1.24 ± 0.08 |
| 30 | 2.01 ± 0.04 |
| 35 | 2.33 ± 0.08 |
| 40 | 3.69 ± 0.09 |
| 45 | 4.69 ± 0.12 |
| 50 | 6.86 ± 0.45 |
| 55 | 8.10 ± 0.59 |
| 60 | 8.62 ± 0.05 |
| 65 | 4.89 ± 0.31 |

Although simultaneous use of Optimash BG and Acremonium cellulase showed the maximum productivity at 60 °C, the rate when incubated at 65 °C was decreased to 57% (Table 1). We considered that this decrease resulted from the inactivation of Acremonium cellulase. Several cellulases from the genus *Acremonium* are inactivated at around 50 °C and lose their activities completely at around 70 °C [15,16].

### 3.2. Application of P. kudriavzevii NBRC1279 and NBRC1664 to SSF for Bioethanol Production

Japanese cedar and eucalyptus show more than 80% of holocellulose yields [17], which releases glucose and other sugars by enzymatic hydrolysis. Thus, Japanese cedar and eucalyptus were used as source materials for SSF.

When the filter paper was hydrolyzed by Optimash BG and Acremonium cellulase, glucose was obtained as the main sugar (Figure 1), which indicated that the combination of both enzymes was effective for glucose production from cellulose and hemicellulose. Thus, we next applied the combination of Optimash BG, Acremonium cellulase, and *P. kudriavzevii* NBRC strains for SSF to produce bioethanol.

When we started this study, the available *P. kudriavzevii* strains were only limited to NBRC1279 and NBRC1664; therefore, both strains were used. Moreover, both strains were capable of producing ethanol in YPD medium with glucose as the sole carbon source (Figure S1, see supplementary data); therefore, *P. kudriavzevii* NBRC1279 and NBRC1664 were applied to SSF. Using Japanese cedar particles as the source material, *P. kudriavzevii* NBRC1279 and NBRC1664 were incubated for 144 h at 35 °C, and the highest ethanol concentrations were reached 21.9 ± 0.50 g/L and 23.8 ± 3.9 g/L, respectively (Figure 2A,B). By contrast, the glucose concentrations when incubated at 35 °C were below the detection limit (Figure 2C,D). Galactose, mannose, arabinose and xylose were also generated, and their amounts were increased as time advanced (Tables S1 and S2), which was caused by carbon catabolite repression [18]. Carbon catabolite repression, a molecular mechanism, suppresses the expression of the enzymes involved in the use of alternate carbon sources in the presence of glucose. Those results indicated that the glucose produced by the enzymatic hydrolysis was consumed as the main carbon source by both strains, and then bioethanol was produced. When both strains were incubated for 144 h at 45 °C, the glucose concentrations were more than 30.0 g/L, but the ethanol concentrations were less than 2.0 g/L. These results indicated that the enzymes had sufficient activities, but that the ethanol productivities of both strains were inert.

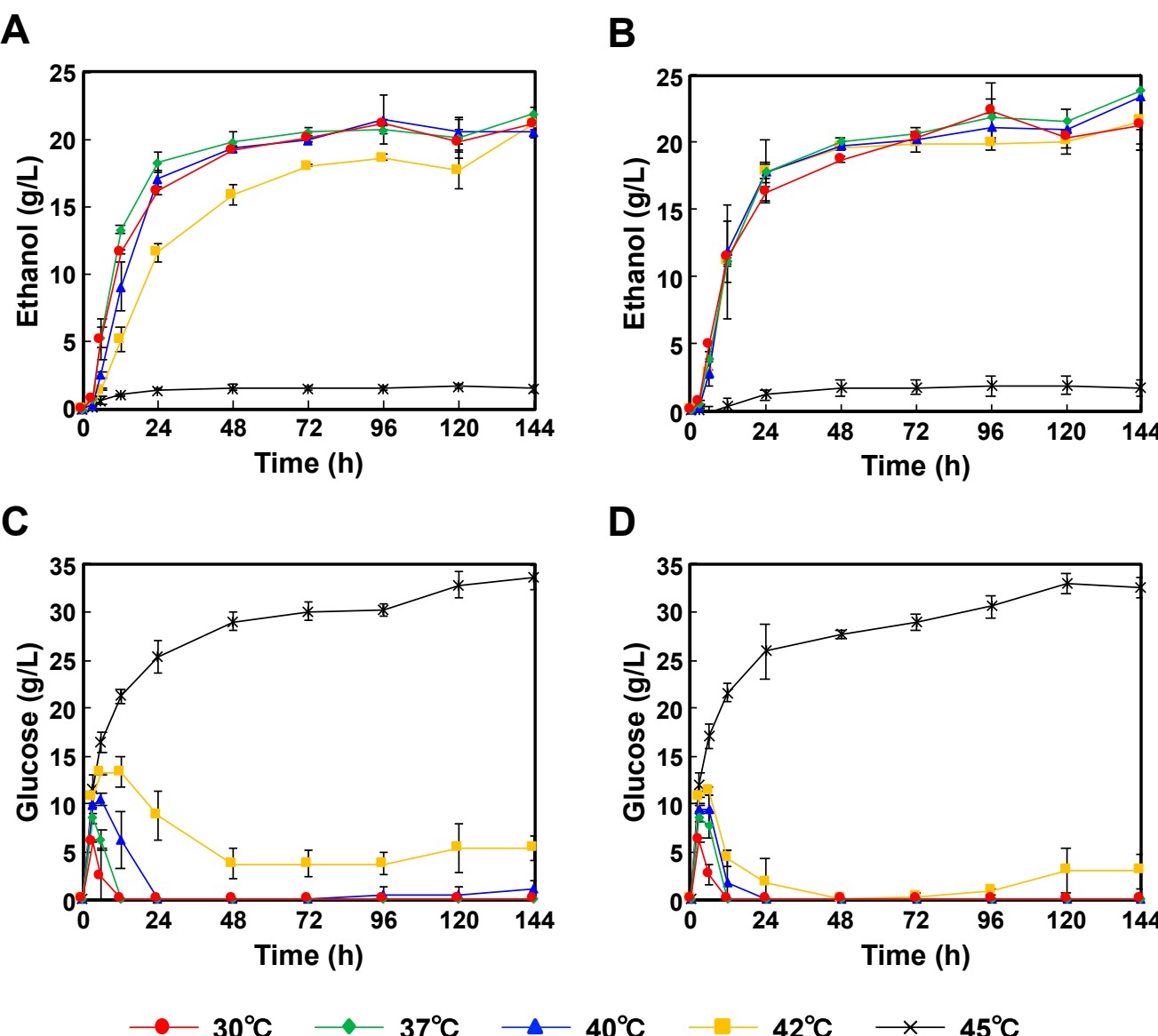

**Figure 2.** Comparison of the time-courses for *P. kudriavzevii* NBRC1279 and NBRC1664 during bioethanol production using Japanese cedar particles by SSF. (**A**) Ethanol production with strain NBRC1279; (**B**) Ethanol production with strain NBRC1664; (**C**) Glucose production with strain NBRC1279; (**D**) Glucose production with strain NBRC1664. Error bars indicate SE (*n* = 3).

When *P. kudriavzevii* NBRC1279 and NBRC1664 were incubated with Japanese eucalyptus particles, the optimum temperatures were 30 °C, and the highest ethanol concentrations were 21.6 ± 0.29 g/L and 21.3 ± 0.21 g/L, respectively (Figure 3A,B). The glucose concentrations were also below the detection limit (Figure 3C,D). As in the case of the SSF using Japanese cedar particles, galactose, mannose, arabinose and xylose were unconsumed (Tables S3 and S4). The profiles of glucose and ethanol concentrations at 45 °C were similar to those using Japanese cedar particles (Figure 2C,D).

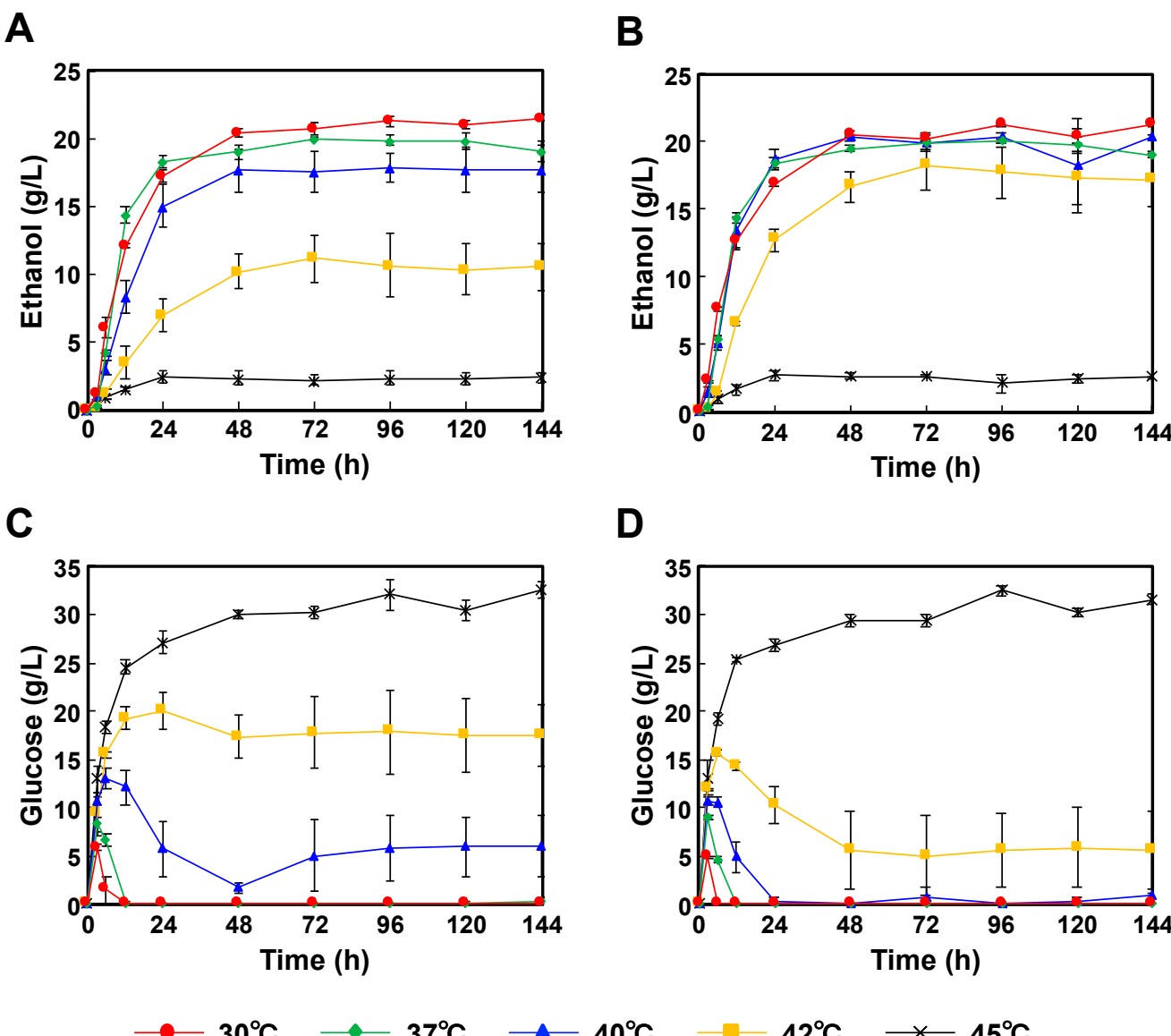

**Figure 3.** Comparison of the time-courses for *P. kudriavzevii* NBRC1279 and NBRC1664 during bioethanol production using Japanese eucalyptus particles by SSF. (**A**) Ethanol production with strain NBRC1279; (**B**) Ethanol production with strain NBRC1664; (**C**) Glucose production with strain NBRC1279; (**D**) Glucose production with strain NBRC1664. Error bars indicate SE (*n* = 3).

When *P. kudriavzevii* NBRC1279 and NBRC1664 were applied to SSF for bioethanol production, both strains showed rapid ethanol productivities in the first 24 h, and then their rates decreased slowly (Figures 2 and 3). When *P. kudriavzevii* NBRC1664 was incubated with Japanese cedar particles at 37–42 °C, more than 17.5 g/L ethanol was produced in 24 h (Figure 2B). However, the ethanol productivity of *P. kudriavzevii* NBRC 1279 was slightly lower than that of strain NBRC1664, and 11.6 g/L ethanol was produced in 24 h at 42 °C (Figure 2A). When Japanese eucalyptus particles were used as the source material, a similar trend in production profiles was observed. After 24 h of incubation at 42 °C, *P. kudriavzevii* NBRC1664 produced 12.7 g/L ethanol, whereas strain NBRC1279 produced 7.0 g/L ethanol (Figure 3A,B). These results demonstrated that *P. kudriavzevii* NBRC1279 and NBRC1664 have similar ethanol productivities at 30–37 °C, but strain NBRC1664 has superior productivity at 40–42 °C. Thus, when comparing the two strains, we considered that strain NBRC1664 has industrial potential for the application of SSF. When the hydrolysate is prepared from Japanese eucalyptus particles, acetate, furfural

and 5-hydroxymethylfurfural are also generated as byproducts [19]. The organic acid and aldehydes inhibit enzymatic activity and microbial growth. In other words, as the temperature increases, the enzyme activity is increased, but the amounts of by-products are also increased. Thus, the ethanol productivity using Japanese eucalyptus particles was lower than that using Japanese cedar particles, and the error bars were larger (Figures 2 and 3). In fact, a similar trend has been observed in the ethanol production using *S. cerevisiae* and *Kluyveromyces marxianus* strains [20].

Several other *P. kudriavzevii* strains have been previously used in SSF for bioethanol production on a laboratory scale (Table 2). At least when comparing the results shown in Table 2, the highest concentrations and productivities in this study were lower than those in previous studies, although it is difficult to accurately compare against results reported previously because the source materials and conditions in SSF are different. However, our method did not require the pretreatment of the source material, such as acid-impregnated steam explosion (AISE) [7], and special expertise for bioethanol production, which could simplify the production process.

**Table 2.** Comparison of bioethanol production.

| Strain | Source Material | Pretreatment Method | Culture Temperature (°C) | Concentratione (g/L) | Productivity [(g/(L·h)] | Reference |
|---|---|---|---|---|---|---|
| *P. kudriavzevii* HOP-1 | Rice straw | Alkali treatment | 40 | 24.3 | 1.10 | [8] |
| *P. kudriavzevii* SI | Rice straw | AISE | 42 | 33.4 | 1.07 | [7] |
| *P. kudriavzevii* NBRC1279 | Japanese cedar | - | 35 | 21.9 | 0.76 * | This study |
| *P. kudriavzevii* NBRC1279 | Japanese eucalyptus | - | 30 | 21.6 | 0.72 * | This study |
| *P. kudriavzevii* NBRC1664 | Japanese cedar | - | 35 | 23.8 | 0.74 * | This study |
| *P. kudriavzevii* NBRC1664 | Japanese eucalyptus | - | 30 | 21.3 | 0.70 * | This study |

* The value was calculated from the concentration after 24 h of incubation.

Using a 70 L scale fermenter, we previously performed the production of bioethanol from hydrolysate prepared from Japanese eucalyptus particles [21]. In this case, the ethanol concentration reached 53.5 g/L, which was equivalent to that observed in laboratory-scale experiments. However, this method was required for the three-step processes of hydrothermal pretreatment (for 2 days), enzymatic hydrolysis (for 3 days), and fermentation (for 3 days). This processing time leads to increased cost. As in the previous method, assuming that the ethanol productivity of SSF is not decreased by the scale-up of the fermenter, our method significantly reduces the processing time. Thus, our method would be much more cost-effective and practical for bioethanol production from lignocellulosic biomass, which may be advantageous for industrial use.

After the enzymatic hydrolysis of filter paper, the maximum concentration of glucose was yielded at 60 °C (Figure 1), and the productivity was more than 80% at 50–60 °C (Table 1). However, the ethanol productivities of *P. kudriavzevii* NBRC1279 and NBRC1664 were significantly reduced when incubated at 45 °C. Thus, it is necessary to resolve the difference between the optimal temperature for enzymatic hydrolysis and the fermentation temperature for bioethanol production for the practical application of SSF. To reduce the difference, we are planning to overexpress CsHSP [22] or HSP20 [23] genes, which enhance heat resistance in *Pichia* cells, in *P. kudriavzevii* NBRC1279 and NBRC1664.

### 3.3. Difference in P. kudriavzevii NBRC1279 and NBRC1664 Based on Draft Genome Sequences

As mentioned above, strain NBRC1664 showed higher ethanol productivities at 40–42 °C than strain NBRC1279 (Figures 2 and 3). To elucidate the differences in ethanol productivity and ethanol tolerance, *P. kudriavzevii* NBRC1279 and NBRC1664 were subjected to genome sequencing, and then genome comparison was carried out.

General genome features of *P. kudriavzevii* NBRC1279 and NBRC1664 are listed in Table 3. Raw data from *P. kudriavzevii* NBRC1279 included 192,636 reads with 201-coverage. The genome sequence was 12,851,201 bases, and the GC content was 37.6%. The assembly

generated 79 contigs with a maximum length of 2,854,910 bases and an N50 contig size of 1,137,449 bases. After the prediction of the open reading frames and proteins, 4897 putative open reading frames were identified in the sequence of strain NBRC1279. We also identified 460 putative tRNA genes. Raw data from *P. kudriavzevii* NBRC1664 yielded 111,234 reads with 102-coverage, and the genome sequence was identified as 12,362,690 bases with a GC content of 37.8%. The genome sequence of *P. kudriavzevii* NBRC1664 was assembled into 76 contigs with a maximum length of 2,539,505 bases and an N50 contig size of 1,482,614 bp. Furthermore, 4815 putative open reading frames were identified.

**Table 3.** Genome features of *P. kudriavzevii* NBRC1279 and NBRC1664.

| Properties | NBRC1279 | NBRC1664 |
| --- | --- | --- |
| Genome length (bp) | 12,851,201 | 12,362,690 |
| GC content (%) | 37.6 | 37.8 |
| Contig numbers | 79 | 76 |
| Coding sequence numbers | 4897 | 4815 |
| tRNA | 460 | 341 |

To clarify the genetic difference between *P. kudriavzevii* NBRC1279 and NBRC1664, we applied the predicted coding sequences from both strains to eggNOG categories and compared them (Table 4). Among these categories, *P. kudriavzevii* NBRC1279 exhibited greatest proportions to function unknown (17.18%); general function prediction only (11.86%); translation, ribosomal structure, and biogenesis (8.54%); posttranslational modification, protein turnover, and chaperones (8.17%); and intracellular trafficking, secretion, and vesicular transport (7.57%). *P. kudriavzevii* NBRC1664 exhibited similar proportions comparable to those of *P. kudriavzevii* NBRC1279 (Table 4).

Subsequently, we constructed the Venn diagram based on the predicted coding sequences in the genomes (Figure 4). The Venn diagram showed 4113 orthologous genes and indicated high similarity between both *P. kudriavzevii* strains. In the Venn diagram, 14 and 7 paralogous genes were present in draft genome sequences of *P. kudriavzevii* NBRC1279 and NBRC1664, respectively (Table 5). The comparison of paralogous genes revealed that five retrotransposon protein genes are contained only within the draft genome of *P. kudriavzevii* NBRC1279. A retrotransposon protein gene can insert into another locus within the genome via an RNA intermediate. For example, the integration of a transposable element in *Schizosaccharomyces pombe* enhances the expression of stress response genes, which leads to improved growth under heat [24], oxidative [24], and heavy metal [25] stresses. Moreover, the upregulation of retrotransposon-related genes enhances ethanol tolerance in *S. cerevisiae* CECT10094 cells [26]. Thus, *P. kudriavzevii* NBRC1279 may have a different capacity of stress tolerances compared with strain NBRC1664. Further work such as transcriptome analysis is required to determine whether an association exists between the retrotransposon proteins and the stress tolerance in strain NBRC1279.

**Table 4.** eggNOG categories of protein coding sequences in *P. kudriavzevii* NBRC1279 and NBRC1664.

| Class | Description | NBRC1279 | | NBRC1664 | |
|---|---|---|---|---|---|
| | | Count | Proportion (%) | Count | Proportion (%) |
| | Information storage and processing | | | | |
| J | Translation, ribosomal structure, and biogenesis | 345 | 8.54 | 349 | 8.85 |
| A | RNA processing and modification | 54 | 1.34 | 50 | 1.27 |
| K | Transcription | 237 | 5.87 | 235 | 5.96 |
| L | Replication, recombination, and repair | 241 | 5.97 | 208 | 5.27 |
| B | Chromatin structure and dynamics | 39 | 0.97 | 40 | 1.01 |
| | Cellular processes and signaling | | | | |
| D | Cell cycle control, cell division, chromosome partitioning | 36 | 0.89 | 38 | 0.96 |
| Y | Nuclear structure | 0 | 0.00 | 0 | 0.00 |
| V | Defense mechanisms | 33 | 0.82 | 37 | 0.94 |
| T | Signal transduction mechanisms | 171 | 4.23 | 176 | 4.46 |
| M | Cell wall/membrane/envelope biogenesis | 45 | 1.11 | 47 | 1.19 |
| N | Cell motility | 7 | 0.17 | 5 | 0.13 |
| Z | Cytoskeleton | 55 | 1.36 | 55 | 1.39 |
| W | Extracellular structures | 0 | 0.00 | 0 | 0.00 |
| U | Intracellular trafficking, secretion, and vesicular transport | 306 | 7.57 | 297 | 7.53 |
| O | Posttranslational modification, protein turnover, chaperones | 330 | 8.17 | 337 | 8.54 |
| | Metabolism | | | | |
| C | Energy production and conversion | 195 | 4.83 | 172 | 4.36 |
| G | Carbohydrate transport and metabolism | 158 | 3.91 | 152 | 3.85 |
| E | Amino acid transport and metabolism | 205 | 5.07 | 205 | 5.20 |
| F | Nucleotide transport and metabolism | 71 | 1.76 | 68 | 1.72 |
| H | Coenzyme transport and metabolism | 93 | 2.30 | 86 | 2.18 |
| I | Lipid transport and metabolism | 104 | 2.57 | 100 | 2.54 |
| P | Inorganic ion transport and metabolism | 99 | 2.45 | 98 | 2.48 |
| Q | Secondary metabolites biosynthesis, transport, and catabolism | 43 | 1.06 | 47 | 1.19 |
| | Poorly characterized | | | | |
| R | General function prediction only | 479 | 11.86 | 469 | 11.89 |
| S | Function unknown | 694 | 17.18 | 673 | 17.06 |

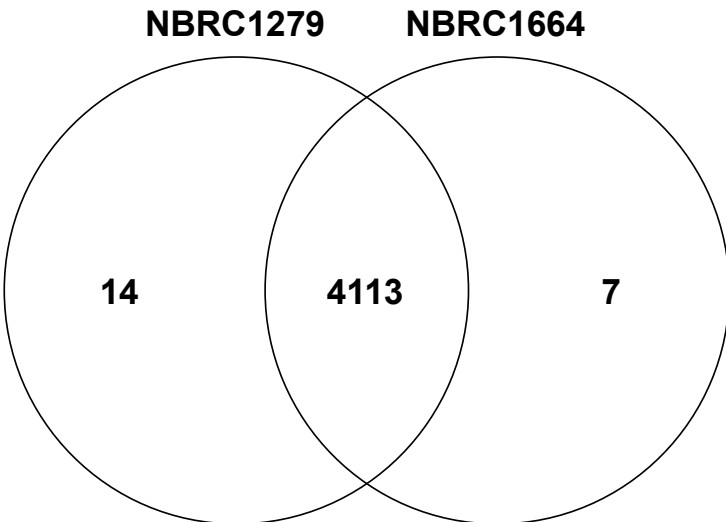

**Figure 4.** The Venn diagram of predicted coding sequences of *P. kudriavzevii* NBRC1279 and NBRC1664.

**Table 5.** Paralogous genes of *P. kudriavzevii* NBRC1279 and NBRC1664.

| NBRC1279 | NBRC1664 |
| --- | --- |
| Aldo/keto reductase family protein | Cytochrome c oxidase subunit 2 |
| Leptin receptor gene-related protein | Hmra1p |
| Molecular chaperones HSP70/HSC70 | Metalloendoprotease HMP1 |
| Monocarboxylate transporter | NADH dehydrogenase subunits 2, 5, and related proteins |
| Peptide methionine sulfoxide reductase | Phenylalanyl-tRNA synthetase, β-subunit |
| Predicted transporter | Phospholipase |
| Retrotransposon protein1 | Uncharacterized conserved protein |
| Retrotransposon protein2 | |
| Retrotransposon protein3 | |
| Retrotransposon protein4 | |
| Retrotransposon protein5 | |
| Solute carrier family 25 | |
| Ubiquitin-conjugating enzyme E2 | |
| Uncharacterized protein | |

Based on the results of the eggNOG categories and the Venn diagram of predicted coding sequences, the enzymes required for ethanol production were similar (Table 4). Thus, we considered that the difference in ethanol productivities was due to the difference in the expression levels of enzymes and enzyme activities related to ethanol production. The draft genome sequences of *P. kudriavzevii* NBRC1279 and NBRC1664 are also useful to identify genes that may confer stress tolerance to microorganisms. In fact, we have previously succeeded in conferring tolerance to acid and salt in *S. cerevisiae* by overexpressing a GPI-anchored protein gene (*IoGAS1*) from *P. kudriavzevii* NBRC1279 [27,28].

*3.4. Genome Properties for Ethanol Tolerance*

To maintain ethanol productivity during SSF, the heat and ethanol tolerances of the host are key factors because high temperature [29] and high concentrations of ethanol [30,31] inhibit the growth of yeast. The thermotolerance mechanism of yeast has been demonstrated by gene expression profiles [29] and proteomics analysis [32,33]. Previous study has also shown that in the bioethanol production from steam-pretreated softwood based on SSF using the *S. cerevisiae* strain, increased ethanol concentration leads to growth inhibition [31]. Therefore, we considered that determining the ethanol tolerance-related genes of *P. kudriavzevii* is important because genetic engineering may be needed to avoid the toxicity of the high concentrations of ethanol produced in the cells [34]. Thus, we tried to confirm

the ethanol tolerance capacities of strains NBRC1279 and NBRC1664 based on the draft genome sequences.

Transcriptome profiling of *P. kudriavzevii* CBS 12547 under ethanol stress has suggested that ethanol tolerance is achieved by various factors [35]. The upregulation of multiple genes involved in ergosterol biosynthesis (*ERG2*, *ERG3*, *ERG27*) and trehalose metabolism (*TPS1*) enhances the accumulation of ergosterol and trehalose, respectively, which is effective for membrane protection of *P. kudriavzevii.* Genes associated with heat stress response (*LRE1*, *WSC1*, *SGT2*) and membrane biogenesis (*RRT12*, *GAS4*, *FLO1*, *IFF6*) have also been upregulated, and these maintain cell wall integrity under ethanol stress. The upregulation of genes associated with molecular chaperones (*HSP42*, *HSP78*, *HSP104*) and the ubiquitin–proteasome system (*UBP16*, *BUL2*, *TOM1*, *HUL4*, *BRE1*, *CUE2*) avoids the aggregation of unfolded or misfolded proteins. In the genome sequence of *P. kudriavzevii* NBRC1279, upregulated genes and homologous genes were conserved. Although *LRE1* was not conserved, other upregulated genes were also conserved in the genome sequence of *P. kudriavzevii* NBRC1664. Thus, we considered that *P. kudriavzevii* NBRC1279 and NBRC1664 have ethanol tolerances similar to that of *P. kudriavzevii* CBS 12547.

## 4. Conclusions

In this study, we confirmed glucose production when using both Optimash BG and Acremonium cellulase. Subsequently, we demonstrated SSF for bioethanol production using *P. kudriavzevii* NBRC1279 and NBRC1664 as well as Optimash BG and Acremonium cellulase. When the highest concentrations and productivities in this study were compared with those of previous studies, those of this study were lower. However, our method did not require pretreatment and special expertise, and is thus much more cost-effective and advantageous for industrial use. To elucidate the differences in the ethanol productivity and ethanol tolerance of both strains, genome sequencing and genome comparison were carried out. The draft genome sequences of *P. kudriavzevii* NBRC1279 and NBRC1664 revealed that both strains have similar ethanol tolerances compared with a related strain. However, further analyses, such as transcriptome analysis, are required to reveal the reasons for the different ethanol productivities between both strains at the genetic level. Based on the Venn diagram, strain NBRC1279 appeared to have a different capacity of stress tolerances from strain NBRC1664.

**Supplementary Materials:** The following are available online at https://www.mdpi.com/article/10.3390/fermentation7020083/s1, Figure S1: Time-dependent changes in ethanol (white circle) and glucose (black square) produced by *P. kudriavzevii* NBRC1279 (A) and NBRC1664 (B). Tables S1 and S2: The other sugar concentration during bioethanol production with *P. kudriavzevii* NBRC1279 and NBRC1664 using Japanese cedar particles. Tables S3 and S4: The other sugar concentration during bioethanol production with *P. kudriavzevii* NBRC1279 and NBRC1664 using Japanese eucalyptus particles.

**Author Contributions:** Conceptualization, H.A., A.M.; methodology, H.A., T.G., Y.I., T.S.; validation, H.A., Y.I.; formal analysis, H.A., T.G.; investigation, H.A., Y.I., Z.-i.K.; resources, A.M.; data curation, H.A., Y.I.; writing—original draft preparation, H.A.; writing—review and editing, H.A., T.G., Y.I., T.S., Z.-i.K., A.M.; visualization, H.A.; supervision, H.A., A.M. All authors have read and agreed to the published version of the manuscript.

**Funding:** This research received no external funding.

**Institutional Review Board Statement:** Not applicable.

**Informed Consent Statement:** Not applicable.

**Data Availability Statement:** Draft genome sequences of *P. kudriavzevii* NBRC1279 and NBRC1664 were deposited in DDBJ/EMBL/GenBank databases under accession numbers BHFO01000001–BHFO01000079 and BHFP01000001–BHFP01000076, respectively.

**Acknowledgments:** We are grateful to all members of the Bio-conversion Research Group at our Institute (Research Institute for Sustainable Chemistry, National Institute of Advanced Industrial Science and Technology (AIST)) for their technical assistance and valuable discussion.

**Conflicts of Interest:** The authors declare no conflict of interest.

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
