# Peer review of "Application of Pichia kudriavzevii NBRC1279 and NBRC1664 to Simultaneous Saccharification and Fermentation for Bioethanol Production"

_fermentation, doi:10.3390/fermentation7020083_

Round 1
Reviewer 1 Report
The work presented by Akita et al., entitled “application of Pichia kudriavzevii NBRC 1279 and NBRC1664 to simultaneous saccharification and fermentation for bioethanol production” describes the production of ethanol with 2 strains of Pichia kudriavzevii. This production from Japanese cedar or eucalyptus particles occurs in a simultaneous step: enzymatic hydrolysis and fermentation. The strategy used and the subject are relevant and interesting mostly when cheap carbon sources are used. However, it cannot be published in the state and some aspects need to be clarified to be published.
ABSTRACT
The abstract is clear and gives a good overview of the work.
INTRODUCTION
Line 48-50: I think you can find other studies with simultaneous saccharification and fermentation without chemical pretreatment step.
METHODS
The methods are not sufficiently described especially the fermentation part.
RESULTS/ DISCUSSION
In figures 1, 2 and 3, there are standard deviations are they obtained by repetitions or with statistical analysis?
I think that to compare the results of fermentation it would be better to use yield mg of ethanol / mg of particle or productivity but in mg of ethanol / mg of particles /min.
Ethanol tolerance is studied but not temperature tolerance why? Is this parameter important?
In a general point of view, the results are not sufficiently commented and discussed, and when they are compared, they are compared with publications which do not seem to be wise: pretreatment. I do not see how these results can be used on an industrial scale.
Author Response
We are grateful to reviewer#1 for the helpful comments and useful suggestions that have helped us to improve our manuscript considerably. As indicated in the responses that follow, we have taken all of these comments and suggestions into account in our revised manuscript. All corrections are highlighted in green color in the revised manuscript. We hope that our revisions are satisfactory.

Reviewer 2 Report
The manuscript presents a study on SSF by two different Pichia Sp on two different substrates. The concerns raised are as follows:
- The basic concern: the biomass composition of the substrates was not analyzed, which is very important to determine hydrolysis efficiency and fermentation yield. Please provide the data.
- The enzyme optimization study and the fermentation study showed that the optimum temperature and pH for enzymatic activity and ethanol fermentation are not close. As can be seen from the fermentation data for both the substrates, a temperature of above 40 degrees is optimal for glucose production, however, at that temperature ethanol production is at its lowest. We cannot even check the efficiency of ethanol production with this substrate, as the authors do not provide positive control with ethanol fermentation of the strains using glucose. Please provide a positive control for ethanol fermentation for both strains under optimal conditions using glucose.
- The pH for saccharification using enzymes is given as pH 5, while the SSF fermentation pH is not given or considered. Why?
- Other sugars of lignocellulosic biomass such as xylose, arabinose are totally ignored. Even the saccharification test has no negative control, and the xylose concentration remains constant throughout the variations in temperature. Pichia sp. can ferment pentoses with other by products. Why is this aspect overlooked?
- In figure 3, the error bars/SD values for fermentations over 40 degrees temperature are so high, while it is not so for figure 2. These are two different substrates, but why one substrate shows such unstable results?
- How can the authors justify the rationale behind genome analysis, while the fermentation dataset is incomplete and some results are inconsistent? Again, fermentation of pentoses are completely ignored.
- Section 3.4 is merely a discussion based on genome data, but not any results from experimental study. Please merge this section with section 3.3.
Author Response
We are grateful to reviewer#2 for reviewing our manuscript. Moreover, your comments and suggestions are very important to improve our manuscript. Our responses to the specific comments and suggestions are described below. All corrections are highlighted in green color in the revised manuscript. We hope that our revisions are satisfactory.

Round 2
Reviewer 2 Report
The authors addressed all my concerns, the manuscript can be accepted in its current form
